# Experimental Determination of the Flow Coefficient for a Constrictor Nozzle with a Critical Outflow of Gas

Victor Bolobov [1], Yana Martynenko [2,*] and Sergey Yurtaev [3]

1   Department of Mechanical Engineering, Faculty of Mechanical Engineering, Saint Petersburg Mining University, 2, 21st Line, Saint Petersburg 199106, Russia; bolobov_vi@pers.spmi.ru
2   Department of Transport and Storage of Oil and Gas, Faculty of Oil and Gas, Saint Petersburg Mining University, 2, 21st Line, Saint Petersburg 199106, Russia
3   Lead Engineer of the Scientific Center "Arctic", Saint Petersburg Mining University, 2, 21st Line, Saint Petersburg 199106, Russia; yurtaev_sl@pers.spmi.ru
*   Correspondence: s205071@stud.spmi.ru; Tel.: +7-9819-594-100

**Abstract:** Reduction of energy expenditures required for various technological processes is a pressing issue in today's economy. One of the ways to solve this issue in regard to liquefied natural gas (LNG) storage is the recovery of its vapours from LNG tanks using an ejector system. In that respect, studies on the outflow of the real gas through the nozzle, the main element of the ejector, and identifying differences from the ideal gas outflow, are of high relevance. Particularly, this concerns the determination of the discharge coefficient μ as the ratio of the actual flowrate to the ideal one, taking into account the energy losses at gas outflow through the nozzle. The discharge coefficient values determined to date for various nozzle geometries are, as a rule, evaluated empirically and contradictory in some cases. The authors suggest determining the discharge coefficient by means of an experiment. This paper includes μ values determined using this method for the critical outflow of air to atmosphere through constrictor nozzles with different outlet diameters (0.003 m; 0.004 m; 0.005 m) in the pressure range at the nozzle inlet of 0.5–0.9 MPa. The obtained results may be used for the design of an ejector system for the recovery of the boil-off gas from LNG tanks, as well as in other fields of industry, for the design of technical and experimental devices with nozzles for various applications.

**Keywords:** discharge coefficient; nozzle; ejector; boil-off gas; critical flow

## 1. Introduction

Based on the analysis of the problems of the energy market [1–3] and the current green energy agenda [4–6], it may be concluded that the issues of waste energy source recovery are among the most pressing. For liquefied natural gas (LNG), one of the possible ways to solve this issue is the recovery and recycling of its boil-off gas, which nowadays is discharged from the LNG tanks using ejectors [7–9]. The use of the ejectors as energy-efficient pumps was the subject of the works of the authors [10–13]; the works of the scientists [14–16] are dedicated to the outflow of gases and liquids from the nozzles of different designs—these are the main components of the ejectors; the papers of the authors [17,18] deal with the calculation of the nozzle geometry and its computer modelling. Nozzles of different configurations are widely used in flowmeters, ejector burners [19], jet engines [20], etc.

It is known [21,22] that during the outflow of the liquid and gas from a constrictor nozzle, the actual gas discharge flow value is smaller than the theoretical one calculated according to gas dynamics formulae. The difference between these flowrates is expressed in the form of a discharge coefficient $\mu$, which is a ratio of an actual flowrate to a theoretical gas flowrate and takes into account the energy losses and jet compression at the outlet section of the nozzle. The data on discharge coefficients and their variations depending on the design features of the device and thermodynamic characteristics of the flow are

necessary for the evaluation of the operating efficiency of the nozzles that have become widely used in the various technical fields mentioned above.

The values of this coefficient for liquids are determined for different nozzle geometries; however, its values for gaseous media are not given in the relevant papers, and if specified, in practice refer to the Laval nozzle only. The fundamental studies—for instance, [23–25]—describing the dynamic properties of inducing jets also do not specify the range of flow losses at critical gas outflow from the nozzle. For instance, the paper [26] on compressed air outflow from the tank provides the following ratio (1):

$$G = \mu f \beta \frac{p}{\sqrt{RT}},$$ (1)

where the coefficient $\mu$ is used without its values indicated. ($f$—nozzle hole section area, $p$—pressure in the tank, $R$—absolute gas constant; $T$—gas temperature, $\beta = \sqrt{k\left(\frac{2}{k+1}\right)^{\frac{k+1}{k-1}}}$, $k$—adiabatic exponent).

In consideration of the process of compressed gas outflow from the tank, the author [27] also refers to the discharge coefficient, which takes into account the flow hydraulic losses at the nozzle outlet without specifying their values and methods of determination.

Among the works where the values of the discharge coefficient for various gas media ($\mu = 0.8$–$0.9$ for argon, helium, hydrogen, nitrogen) are specified, the works [28,29] should be noted; however, these focus on the gas outflow from the Laval nozzle only.

The authors of this article were able to obtain certain data on $\mu$ values for gas media, using an example of the boil-off fraction of liquefied natural gas discharged at a critical rate through a safety valve from the LNG storage tank [30], by analysing the technical operating conditions of the specified valve [31] and comparing the specified values of the effective $f_{ef}$ and nominal $f_n$ cross-sectional areas of the valve seat, as $\mu = f_{ef}/f_n$. The physical concept of the discharge coefficient for gas [31] is defined as the ratio of the gas mass flow through the valve to the gas flow through the ideal nozzle with an opening area equal to the smallest section area of the valve seat.

The calculated $\mu$ values, depending on the pressure of the medium upstream the valve $p^{inlet}$ and the nominal diameter of the inlet nozzle $d$, are shown in Table 1:

**Table 1.** Discharge coefficient for the LNG tank pressure safety valve.

| $p^{inlet}$, **MPa** | $d$, **m** | $f_{ef}$, **m$^2$** | $\mu$ |
|---|---|---|---|
| 6 | 0.05 | $0.12 \times 10^{-2}$ | 0.61 |
| 6 | 0.2 | 0.016 | 0.51 |
| 16 | 0.05 | $0.46 \times 10^{-3}$ | 0.23 |
| 16 | 0.2 | $0.64 \times 10^{-2}$ | 0.2 |

Based on the represented data, it may be concluded that the value of $\mu$ can vary over a wide range, depending on the constrictor nozzle design and the gas discharge conditions, and the lack of adequate information about the actual flowrate can lead to emergency situations. Primarily, this concerns tank storage facilities for compressed gases and cryogenic products, which include LNG.

As for the calculation of the $\mu$ values, the amount of relevant information has proven to be very limited.

For instance, according to [21,32], the nozzle discharge coefficient in the process of the medium outflow can be represented as a product of (2):

$$\mu = \varepsilon \cdot \varphi$$ (2)

where $\varepsilon = F_{ef}/F_n$ is a jet compression factor ($F_{ef}$—area of contracted cross-section of the jet; $F_n$—area of the nozzle opening); $\varphi$—velocity coefficient determined experimentally by

calculation as a ratio of the experimentally determined flow velocity in the nozzle section $V_{exp}$ to the theoretical one $V_{ideal}$ (3):

$$\varphi = \frac{V_{\text{exp}}}{V_{ideal}},\qquad(3)$$

or calculated as $\varphi = 1/\sqrt{\alpha_c + \xi}$ ($\alpha_c$—Coriolis coefficient; $\xi$—local resistance coefficient). Moreover, [33] indicates that the calculation of each of the multipliers (2) presents some difficulties.

In [32,34], the value of the discharge coefficient is associated with the relative length of the nozzle $l/d$ ($l$—the length of the constrictor section of a nozzle; $d$—diameter of the nozzle outlet section); however, it is clarified that $\mu$ is determined by experiment.

In the article [35], it has been observed that the gas-dynamic losses in the nozzles of various designs increase as the nozzle section area $F_{cr}$, pressure $p$ and taper angle $\alpha$, i.e., $\mu{\sim}1/F_{cr}$; $\mu{\sim}1/p$; $\mu{\sim}1/\alpha$ increase. At the same time, the dependence of $\mu$ from the length of the cylindrical section of the nozzle is of a different nature—$\mu{\sim}l$.

According to [36], the values of the discharge coefficient for gas media are always smaller than for the liquid ones, which is due to the difference in the media viscosity and, therefore, in the media flow mode.

The article [37] mentions the reduction of the flow coefficient as the pressure drop downstream and upstream the nozzle increases and provides experimental data suggesting that when the pressure ratio changes from 1.1 MPa/0.1 MPa to 0.4 MPa/0.1 MPa, the value of the discharge coefficient increases by 1.5–2 times.

The aim of this study, using air as an example, was to develop a methodology to determine the discharge coefficient by experimental calculation and use it to establish the discharge coefficient for the critical gas outflow from constrictor nozzles of different internal diameters.

## 2. Materials and Methods

Gas outflow was carried out into the environment from a constant volume tank ($V = 0.1$ m$^3$) filled with compressed air ($p_0^{\tan k} = 1.05$ MPa; $T_0 = 293$ K) and equipped with a reducer set to the pressure of $p_0^{inlet} = 0.9$ MPa; 0.7 MPa; 0.5 MPa through the constrictor nozzles of diameter $d_1 = 0.003$ m; $d_2 = 0.004$ m; $d_3 = 0.005$ m taper angle $\alpha = 15°$ (Figure 1). At any specific moment of the outflow $t_i$, pressure in the tank $p_i^{\tan k}$ and at the nozzle inlet downstream the reducer $p_i^{inlet}$ was measured. The purpose of the experiment was to construct a time dependence of the mass flowrate at the outflow of air from the tank.

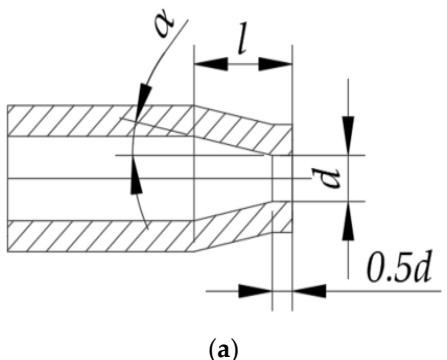

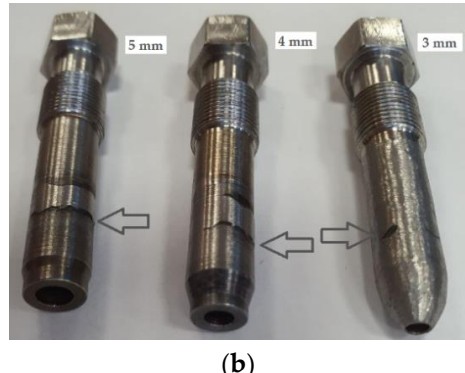

|   (a)   |   (b)   |

**Figure 1.** A sketch (**a**) and an exterior (**b**) of the nozzle with a welded supply tube. (arrow—welding spot) [compiled by the author Y. V. Martynenko].

The equipment of the test bench (drawing 2):
- reciprocating compressor 1 REMEZA СБ4/С-100.LB30 (pressure 1.05 MPa; receiver 2 of 0.1 m$^3$ capacity; pressure gauge 3 of accuracy class 1 with an error of 0.016 MPa);

- ball valves 4, 8 with a nominal bore of 6 mm matching the diameter of the supply tube to the nozzle;
- reducer 5 with a nominal bore of 6 mm, equipped with a manometer of accuracy class 1 with an error of 0.01 MPa;
- electronic pressure transmitter 6 with an accuracy of ±1% FS, transmitting data to a personal computer with a frequency of 1 s;
- removable constrictor nozzles with a threaded connection 9.

Sequence of operations on the test bench (Figure 2):

1.  Nozzle 9 under test is to be fixed to the compressed air line 6 using a threaded connection;
2.  The ball valve on the compressed air line 4 is closed. Compressor 1 supplies air into the receiver up to an excess pressure of 1 MPa (maximum compressor pressure possible), which is checked visually using a pressure gauge 3;
3.  The pressure reducer 5 is adjusted to the required pressure (valve 4 is open and valve 8 is closed);
4.  Valve 8 is switched to an "open" position, and air flows through the constrictor nozzle 9 into the atmosphere. The pressure drop in the receiver is checked visually using a pressure gauge 3. The pressure transmitter 6 transmits the readings of the reduced flow to the computer 11;
5.  The operation is repeated with nozzles of variable diameters and pressure reducer adjusted to pressures ensuring the critical drop at discharge to atmosphere.

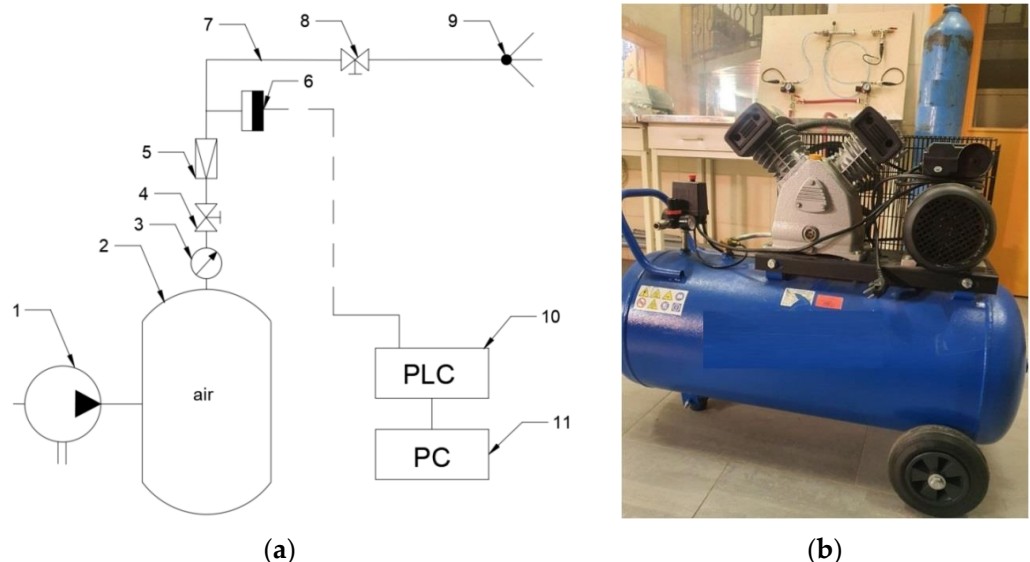

(**a**)                                                                                              (**b**)

**Figure 2.** A sketch (**a**) and an exterior (**b**) of the test bench [compiled by the authors Y.V. Martynenko, V.I. Bolobov]: 1—compressor; 2—receiver; 3—pressure gauge at receiver; 4, 8—ball valve; 5—reducer; 6—pressure transmitter; 7—compressed air supply line; 9—constrictor nozzle; 10—programmable logic controller; 11—personal computer.

Due to the difficulty of recording the real temperature of the gas medium in the receiver, the calculation of the gas mass flowrate at the receiver emptying shall be performed under two assumptions:

- here is practically no heat exchange between the gas in the receiver and the ambient environment; therefore, the process of the receiver emptying may be regarded as adiabatic;
- as a result of intensive heat exchange with the environment, the gas inside the receiver maintains its initial temperature (ambient temperature) at any moment of emptying, i.e., the emptying process is isothermal.

The calculation for both cases has been performed under the assumption that the thermophysical properties of the analysed gas are close to those of an ideal gas.

In the case of the *adiabatic* process (option 1):

From the ratio $p_i^{\tan k} / p_0^{\tan k} = (m_i/m_0)^k$ linking the pressure and mass of the gas in the tank at each *i*-th and initial instant, it follows that (4):

$$m_i = e^{\frac{\ln \frac{p_i^{\tan k}}{p_0^{\tan k}} + k \ln m_0}{k}}.\tag{4}$$

At that, $m_0 = p_0^{\tan k} V \rho_{st}$, where $\rho_{st}$ is the air density at standard conditions ($T^{st}$ = 293 K; $p^{st}$ = 0.1 MPa).

At that, the mean value of the mass flow of gas from the tank for each time interval under consideration in the experiment is as follows (5):

$$G_{\exp} = m_i - m_{i+1}/t_{i+1} - t_i.\tag{5}$$

Under the assumption that the receiver emptying is isothermal (option 2), the formula (6) is valid for the flowrate calculation:

$$G_{\prime\exp} = \frac{V \cdot (p_i - p_{i+1})}{t_{i+1} - t_i} \rho_{st}.\tag{6}$$

The determined values $G^{exp}$ were compared with the values of this indicator calculated by the Equation (1) for the critical outflow of air ($k$ = 1.41; $R$ = 287.1 J/kg·K) through the nozzle at an inlet pressure $p_i^{inlet}$, assuming that the gas outflow is ideal, with no flow resistance and energy losses along the length of the nozzle ($\mu$ = 1). The gas temperature $T_i$ in the Formula (1) was determined based on the ratio (7) [38]:

$$T_i = T_0 \left( \frac{p_i^{inlet}}{p_0^{inlet}} \right)^{\frac{k-1}{k}}.\tag{7}$$

It can be noted that, owing to compliance with the conditions for a critical pressure drop at the nozzle inlet and outlet in the experiment ($p_{atm}/p_{inlet}$ < 0.528 necessary pressure ratio for the critical pressure drop), the Formula (1) was considered valid for the entire analysed time interval.

## 3. Results and Discussion

The results of calculating (5) the mean values of the gas mass flow at the tank emptying $G_{exp}$ for each of the analysed time intervals for different pressures across the reducer $p_0^{inlet}$ and nozzle diameters $d$ for the adiabatic conditions of the receiver emptying are shown as graphs in Figure 3 (graphs for isothermal conditions had a similar form and, therefore, are not shown). It also includes the time dependences of the air mass flow $G_{ideal}$, calculated (1) for the ideal conditions of gas outflow from the nozzle (at $\mu$ = 1).

As seen in Figure 3, for all values $p_0^{inlet}$ and $d$, gas discharge curves at its outflow from the nozzle under the ideal conditions are above those determined by the real change in the gas mass in the receiver, which allows calculation of the actual flow coefficient of the used nozzles. Figure 4 shows the arithmetic average of all values of the ratio $\mu = G_{\exp}/G_{ideal}$ for the entire time interval analysed in the experiments for two heat exchange options between the gas in the receiver and the environment (adiabatic and isothermal conditions).

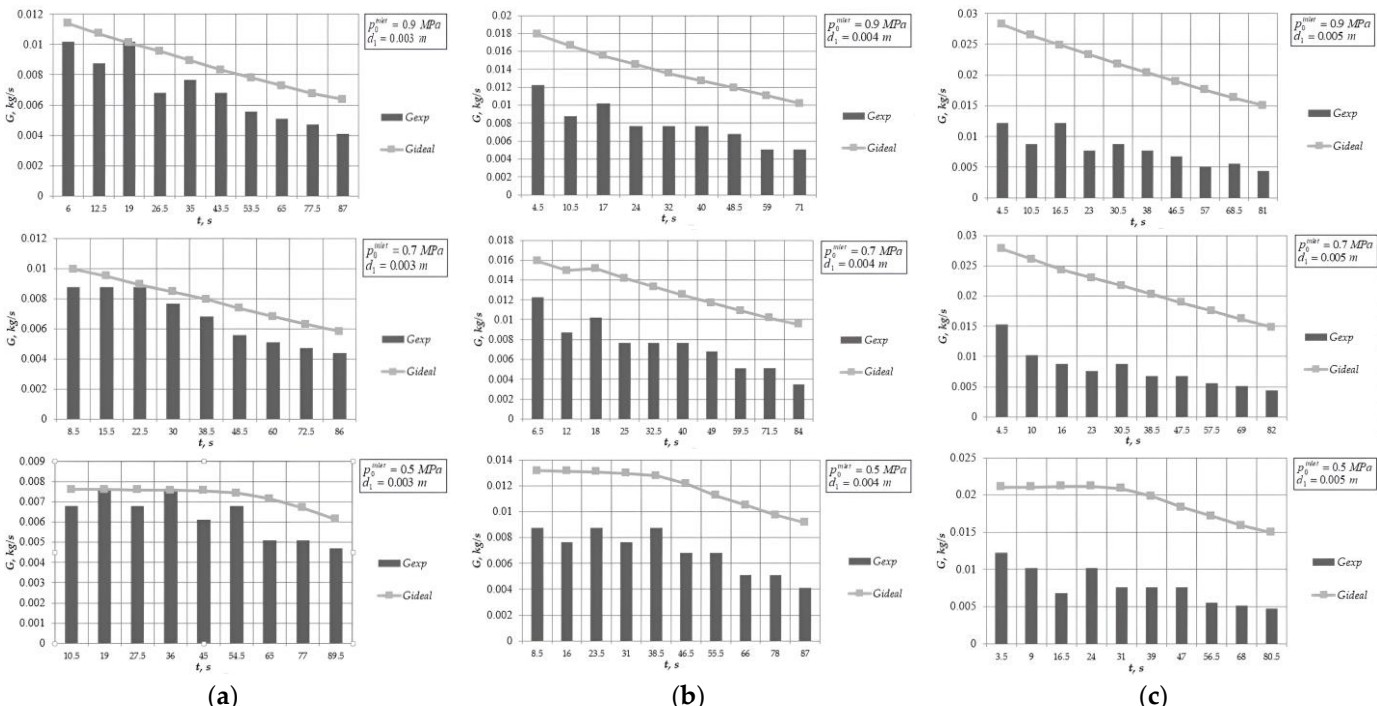

**Figure 3.** Air flow time dependences calculated based on the change in the mass of gas in the receiver (graphs) and the gas outflow through the nozzle under ideal conditions (curves) using nozzles with of diameter d = 3 mm (**a**), 4 mm (**b**), 5 mm (**c**) [compiled by the authors V. Bolobov, Y. Martynenko].

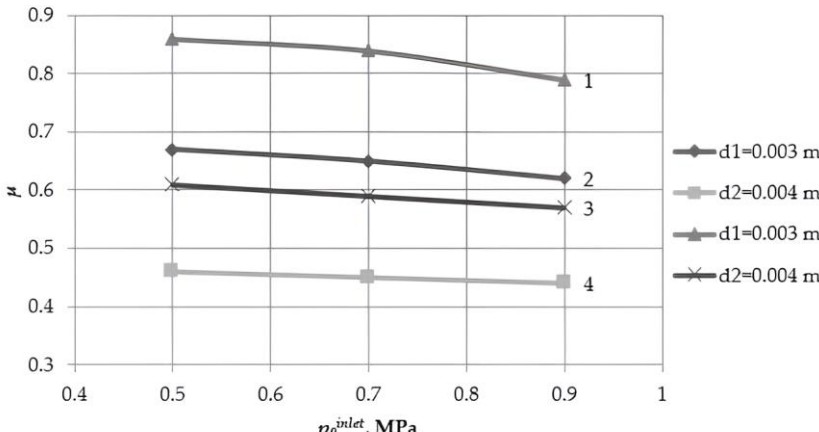

**Figure 4.** Values of a discharge coefficient of a constrictor nozzle based on the inlet pressure upstream the nozzle, calculated for the isothermal (curves 1, 3) and adiabatic (curves 2, 4) conditions of the receiver emptying [compiled by the authors V. Bolobov, Y. Martynenko].

It was found (Figure 4), for the conditions of the experiment, the values of the discharge coefficient $\mu$ are within the range of 0.86–0.44 and decrease as the gas pressure upstream the nozzle and the nozzle diameter increase. It can be noted that the same trend of $\mu$ decrease is observed with the gas outflow through the safety valve (Table 1), as well as in the experiment results of the authors [35,37,39].

The specified decrease in the $\mu$ values (increase of losses in the nozzle) with an increase of $p$ and $d$ stem from the fact that when the gas flows through the constrictor nozzle, discharge zones are formed in the near-wall areas, the sizes of which increase with an increase in the pressure difference upstream and downstream the nozzle [40] and also, possibly, with an increase in the nozzle diameter. Their presence narrows down the effective area of the gas outflow, which is expressed as a decrease in the flow coefficient.

As seen in Figure 4, the expression $\mu = f(p)$ for all analysed nozzle diameters, regardless of the assumptions made and whether the receiver emptying is adiabatic or isothermal, is satisfactorily approximated by the following linear expression:

$$\mu = \mu_0 - K_1 p, \tag{8}$$

with the coefficient $K_1$ averaged for the two considered heat exchange modes equal to 0.113 MPa$^{-1}$ and $\mu_0$ decreasing as the nozzle diameter increases.

Using the linear interpolation of the dependency $\mu_0 = f(d)$, the expression is as follows

$$\mu_0 = \mu_0^* - K_2 d, \tag{9}$$

with the averaged values $\mu_0^*$, $K_2$ for the two heat exchange modes equal to 1.62 and 263 m$^{-1}$, respectively.

Substitution of the expression (9) into (8) results in Equation (10), which allows us to calculate the air discharge coefficient depending on its pressure $p$ upstream the constrictor nozzle and diameter $d$ of the nozzle:

$$\mu_0 = 1.62 - 263p - 0.113d, \tag{10}$$

where $p$ is given in MPa, and $d$ is in m.

It may be noted that low discharge coefficient values obtained in this work (0.86–0.44) indicate high energy losses in the nozzles under consideration. It could be assumed that the said losses are caused by considerable roughness of the inner surface of the nozzle walls and the presence of metal overlaps (dark spots in Figure 5a), which appeared in the process of welding the nozzle to the supply tube. However, replicated experiments carried out on nozzles with a machined inner surface (Figure 5b) did not show a noticeable increase in the discharge coefficient, which indicates an insignificant influence of the condition of the nozzle inner surface on the energy losses of the flow in the nozzle.

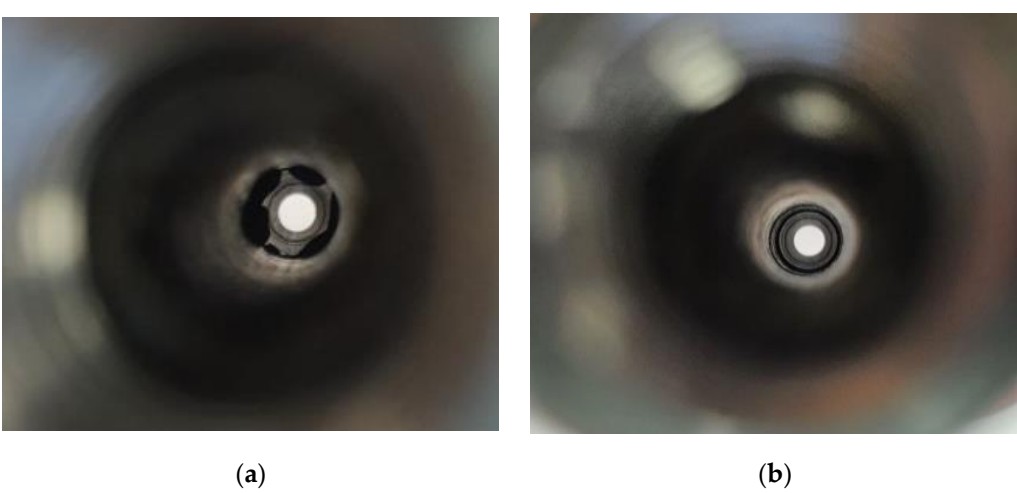

(**a**)          (**b**)

**Figure 5.** View of the inner surface of the nozzle before (**a**) and after (**b**) machining [compiled by the author S. Yurtaev].

Thus, it can be assumed that the obtained low $\mu$ values are associated with the design features of the analysed nozzles, among which, primarily, could be insufficient lengths of the constrictor and cylindrical parts.

Nevertheless, the performed analysis shows that the obtained discharge coefficient values are comparable with the ones obtained by other authors for the outflow of the liquid (0.82 [33]; 0.7–0.95 for $p_0^{inlet}$ from 3 to 12 MPa [41]; 0.7–0.85 $p_0^{inlet}$ to 0.15 MPa [22]) and gas [29] by experiment (0.8–0.95) and calculation (0.7–0.95).

## 4. Conclusions

The hydraulic calculation of nozzles, openings and safety valves cannot be carried out without the discharge coefficient, which accounts for the energy losses of the flow passing through the section of these components. The discharge coefficient is an empirical value that depends on many factors, including the density of the measured medium, viscosity, flowrate, geometric dimensions and roughness [42]. Thus, the coefficient values obtained from a theoretical calculation provide a result close to real conditions [29]. There is a sufficient amount of experimental data on the outflow of liquid; however, the data on the discharge coefficient for gas media from the constrictor nozzles are usually not provided or apply to the Laval nozzle only.

The paper suggests an analytical–experimental method of determining the discharge coefficient by taking pressure readings at different time intervals in a tank with compressed air, from which the gas is fed to the nozzle and at the nozzle, followed by correlation of the obtained mass flowrates. Assuming that the process of vessel emptying is close to adiabatic or isothermal, and the gas outflows through the nozzle with no heat exchange with the ambient environment, i.e., under adiabatic conditions, the discharge coefficients μ for three nozzles of different diameters, representing a combination of constrictor and cylindrical parts, have been calculated. It has been found that for the given nozzle design, the values of the coefficient μ are in the range of 0.86–0.44 and decrease as the gas pressure upstream the nozzle and the nozzle diameter increase, which confirms the patterns discovered by other authors.

**Author Contributions:** Conceptualization, V.B. and Y.M.; methodology, V.B.; formal analysis, Y.M.; investigation, Y.M.; resources, S.Y.; data curation, S.Y.; writing—original draft preparation, Y.M.; writing—review and editing, V.B.; visualization, Y.M.; supervision, V.B.; project administration, S.Y. All authors have read and agreed to the published version of the manuscript.

**Funding:** This research received no external funding.

**Data Availability Statement:** The data presented in this study are available in patent «Method of utilizationof steam gas from a liquefied natural gas (LNG) tank», No. 2770964 at the Federal Institute of Intellectual Property of the Russian Federation and article https://doi.org/10.3390/inventions7010014.

**Acknowledgments:** The authors are grateful to the Arctic Research Center and the Department of Oil and Gas Transportation and Storage of St. Petersburg Mining University for the opportunity to conduct experiments.

**Conflicts of Interest:** The authors declare no conflict of interest.

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
