# Peer review of "Experimental Determination of the Flow Coefficient for a Constrictor Nozzle with a Critical Outflow of Gas"

_fluids, doi:10.3390/fluids8060169_

Round 1

Reviewer 1 Report

The topic of the manuscript “Experimental determination of the flow coefficient for a constrictor nozzle with a critical outflow of gas” is interesting and the manuscript is well written. However, the fundamental of the approach and the results need to be improved.

  • Some grammatical errors. For example,  the style “”The author [25]” needs to be changed. On Line 60: the unit m3/s should be improved.  On line 152, the range. Similarly on line 169 and 170. 
  • The symbol should be introduced when it was used at the first time. For example, “l” on line 90.  
  • Since the experimental observations were make at a very few point, is it possible to run get more tests? Especially, for the conclusion that the discharge coefficient will be used for without fine machining of the surface. How and the validation is vague.   More experimental data need to support such a conclusion. 
  • More important, how can the results for these experiments be extended to other design? More explanation of the validation is needed.  

Author Response

Dear Reviewer,
Thank you for your feedback on our manuscript. Thanks to your recommendations, we have been able to improve and revise our previous studies. We hope that the work will receive your positive feedback.

Reviewer 2 Report

The paper titled "Experimental determination of the flow coefficient for a constrictor nozzle with a critical outflow of gas" presents experimental results on the flow coefficient of a constrictor nozzle with a critical outflow of gas. While the study presents some interesting findings, there are several areas that need to be improved. The following observations and recommendations are suggested:

 ·       The introduction section failed to discuss the importance, research gap, and justification for performing this study. The author should provide a more thorough literature review to give the reader a better understanding of the background and significance of the research problem. The introduction should also discuss the research gap and justification for performing the study.

·       The Materials and Methods section needs to include a figure of the test rig and detailed specifications of the measuring equipment used in the experiment. Additionally, the author should provide information on the uncertainty associated with the measuring equipment. The process of experimentation should also be included, outlining the steps taken in the experiment.

·       The Results section needs a more detailed discussion of the findings. The author should describe the trends and patterns observed in the data and explain the significance of the results. The section should also include statistical analyses, including measures of central tendency and variability, as appropriate.

·       The author needs to discuss why the experimental results mentioned in Figure 2 are inconsistent. The cause of the inconsistency should be investigated, and possible sources of error should be identified.

·       The Conclusion section should be based only on the author's findings. The author should summarize the key findings of the study and explain their significance. The conclusion should also highlight the limitations of the study and suggest areas for future research.

Author Response

(The authors gave the same response as above.)

Reviewer 3 Report

1. The English should be improved.

2. Only two nozzles with different dimensions were tested in this study. Too few experiments to infer laws.

3. The analysis about the experimental data is too simple, and the application scope of the discharge coefficient is unclear.

4. The calculation correlations of discharge coefficient is suggested to added.

Author Response

(The authors gave the same response as above.)

Round 2

Reviewer 2 Report

As the reviewer for this manuscript, I am pleased to report that the authors have made significant improvements based on the feedback and suggestions provided. They have carefully addressed all the comments and incorporated them into the revised manuscript, which has resulted in a much more comprehensive and detailed study.

Reviewer 3 Report

The paper is suggested to be accepted.